# Antidiabetic Activities and GC-MS Analysis of 4-Methoxychalcone

**Leonard D. R. Acho** [1], **Edinilze S. C. Oliveira** [2], **Simone B. Carneiro** [1], **Fernanda Paula A. Melo** [3], **Leilane de S. Mendonça** [1], **Renyer A. Costa** [2], **Rosivaldo S. Borges** [3], **Marcos B. Machado** [2], **Hector H. F. Koolen** [4], **Igor Rafael dos S. Magalhães** [1] and **Emersom S. Lima** [1,*]

1. College of Pharmaceutical Sciences, Federal University of Amazonas, Manaus 69077-000, AM, Brazil; tigrefarma84@hotmail.com (L.D.R.A.); braga.simone.c@gmail.com (S.B.C.); leilane.smendonca@gmail.com (L.d.S.M.); magalhaes.irs@gmail.com (I.R.d.S.M.)
2. Department of Chemistry, Institute of Exact Sciences, Federal University of Amazonas, Manaus 69077-000, AM, Brazil; edinilzeoliveira@gmail.com (E.S.C.O.); renyer.costa@gmail.com (R.A.C.); marcosmachado@ufam.edu.br (M.B.M.)
3. Center for Studies and Selection of Bioactive Molecules, Institute of Health Sciences, Federal University of Pará, Belém 66075-110, PA, Brazil; fernandapamelo@gmail.com (F.P.A.M.); rosborg@ufpa.br (R.S.B.)
4. Research Group in Metabolomics and Mass Spectrometry, School of Health Sciences, State University of Amazonas, Manaus 69005-010, AM, Brazil; hectorkoolen@gmail.com
* Correspondence: eslima@ufam.edu.br

**Abstract:** Diabetes mellitus is a chronic metabolic disease that is mainly characterized by hyperglycemia. Chalcones and their derivatives have demonstrated promising pharmacological potential for the treatment of diabetes. The aim of the study was to evaluate antidiabetic activities and analyze 4-methoxychalcone (MPP) using GC-MS. The compound was characterized using mass spectroscopy, nuclear magnetic resonance and headspace with gas chromatography coupled to mass spectrometry (HS-GC-MS). MPP was evaluated via the inhibition of the alpha-glucosidase enzyme, cell viability and antiglycation and hemolytic activities in vitro. The study of the interaction between the bovine serum albumin protein and MPP was investigated via molecular docking. Oral sucrose tolerance and oral glucose tolerance tests were performed in streptozotocin (STZ)-induced diabetic mice. The HS-GC-MS method was able to accurately detect and characterize the compound, and the interaction between MPP and BSA revealed the remarkable affinity for the two main binding sites of BSA. This was confirmed by the in vitro antiglycation test, since MPP showed activity through both oxidative and non-oxidative stress. MPP significantly attenuated the increase in glycemia after glucose loading in STZ-induced diabetic mice. These results confirm that MPP has antihyperglycemic activity and may be an alternative for the treatment of diabetes mellitus.

**Keywords:** 4-methoxychalcone; diabetes; antiglycant; antihyperglycemic

## 1. Introduction

Diabetes mellitus (DM) is a set of metabolic disorders that is characterized by increased blood glucose, and it has been considered to be one of the biggest current health problems in the world [1]. In all countries, there has been a significant increase in diabetes in recent years [1]. The overall prevalence of DM in adults increases with age, and there is a higher prevalence in men than in women [2]. In 2021 alone, excluding the mortality risks associated with the COVID-19 pandemic, an estimated 6.7 million adults between the ages of 20 and 79 died from the consequences of diabetes [3]. Future projections suggest that 783 million people will have diabetes by 2045 [2].

The development of sequelae as a result of DM has been associated with the accumulation of advanced glycation end products (AGEs), which are proteins or glycated lipids that form as a result of chronic hyperglycemia [4,5]. AGEs are related to the multiple complications of diabetes [4], such as retinopathy, renal failure and amputation [6], in addition to

hypertension [7], decreased muscle mass and performance [8], decreased bone density and strength [8,9] and the activation of inflammatory cytokines [10]. Some existing therapies for the treatment of DM make use of inhibitors of the enzyme $\alpha$-glucosidase, such as acarbose and voglibose, which are commonly prescribed to lower postprandial glucose. However, this does not prevent the long-term complications of diabetes, in addition to being able to cause unwanted side effects such as nausea, vomiting and abdominal discomfort [11,12]. Although metformin, which is an oral hypoglycemic agent, has become an option for the treatment of type 2 DM, some patients cannot receive it due to the risk of lactic acidosis [13]. These effects, added to low adherence to diabetic treatment, contribute to the substantial aggravation of this disease, thus increasing treatment costs and sometimes leading to death [14].

Given this context, in order to enable the emergence of new classes of more effective drugs, new alternatives with fewer side effects and lower cost have been developed [15,16]. Chalcones, which belong to the class of flavonoids, have attracted the attention of researchers not only from a synthetic and biosynthetic point of view but also from a pharmacological point of view [17]. Claisen–Schmidt condensation is typically used to synthesize chalcones in the presence of basic catalysts [18,19]. These substances have already revealed biological properties for the treatment of diabetes [18,20–22] and other actions that include antimutagenic [23], anti-inflammatory [24,25], antimicrobial [26], antifungal [27] and antiproliferative [28] activities.

Therefore, this study addresses the anti-diabetic potential was analyzed in in vitro and in vivo experimental models of the 4-methoxychalcone. For the evaluation of its antiglycation activity in vitro and its ability to interact with bovine serum albumin, in silico and chemical characterization using the headspace technique with gas chromatography coupled to mass spectrometry were performed.

## 2. Materials and Methods

### 2.1. Materials

Intestinal acetone powders from rat, $\alpha$-glucosidase of *Saccharomyces cerevisiae*, acarbose, 4-nitrophenyl $\alpha$-D-glucopyranoside, resazurin sodium salt, penicillin-streptomycin, streptozotocin (STZ), glyoxal, phosphate-buffered saline (PBS), bovine serum albumin (BSA), aminoguanidine and fructose were purchased from Sigma-Aldrich (St Louis, MO, USA). The MRC-5 cell line (human lung fibroblast culture) was obtained from the Cell Bank of Rio de Janeiro (Brazil). Methanol and dimethyl sulfoxide (DMSO) were purchased from Tedia (Fairfield, OH, USA). The deuterated solvent dimethylsulfoxide with tetramethylsilane (TMS, D, 98.0%) was purchased from Cambridge Isotope Laboratories Inc. (Andover, MA, USA). The reaction was monitored using thin layer chromatography (TLC) and an F-254 silica gel glass plate (20 cm $\times$ 20 cm). The melting point measurement was determined in an electrothermal melting point apparatus (MQAPF-302, Microquímica Equipamentos Ltd.a, Palhoça, SC, Brazil).

### 2.2. Synthesis

The synthesis of 4-methoxychalcone or [(*E*)-3-(4-methoxyphenyl)-1-phenylprop-2-en-1-one] was carried out using the Claisen-Schmidt condensation reaction. A mixture of 6.0 mL of acetophenone (50.0 mmol) and 6.56 mL of *p*-methoxy-benzaldehyde (55.0 mmol) was dissolved in 15 mL of ethanol, which was cooled in an ice bath in order to add 25 mL of an aqueous solution of sodium hydroxide (63.0 mmol). Subsequently, the reaction mixture was kept under constant magnetic stirring for 2 h, at a temperature between 25–30 °C, and then kept for 24 h under refrigeration. The crystals were separated after washing with chilled distilled water and vacuum filtration. The crude product was recrystallized using hot ethanol, and a yield of 92.0% of 4-methoxychalcone was obtained.

## 2.3. Characterization by NMR and ESI-MS/MS

The NMR analyses ([1]H, [13]C, DEPT135, [1]H-[1]H COSY, HSQC and HMBC) of the MPP were performed using an NMR spectrometer [(Bruker[®®] Avance III HD) at 500.13 MHz for protons and 125.8 MHz for [13]C, BBFO Plus SmartProbe[TM] (New York, NY, USA)] at 298.0 K. The [1]H NMR spectra were acquired using the ZGPR (residual water signal suppression) pulse sequence. The chemical displacements are presented in ppm and have the TMS (0.0 ppm) as the internal reference. For the characterization of the chemical structure using NMR, the following abbreviations are used: *s* = simplet, *d* = doublet, *m* = multiplet and *J* (coupling constant in Hz). To acquire the mass spectra, a quadrupole triple mass spectrometer (TSQ Quantum Access, Thermo Scientific[TM], San Jose, CA, USA) was used. The sample was solubilized in chloroform (1 mg mL$^{-1}$) and subsequently diluted to 5 μg mL$^{-1}$. The electrospray ionization source (ESI) was used in positive mode for the ESI-MS and ESI-MS/MS analyses by direct infusion (5 μL). The analytical conditions were as follows: acquisition range *m*/*z* 100–1000 Da, capillary potential of 40 V; spray voltage of 8 kV; capillary temperature 247 °C; drag gas $N_2$, flow of 11 arbitrary units (arb). Fragmentation was performed with collision energy between 35 to 45 eV. The Xcalibur[TM] software version 2.2 (Thermo Scientific[TM]) was used during the acquisition and processing of the spectra.

(*E*)-3-(4-methoxyphenyl)-1-phenylprop-2-en-1-one (MPP): yellow powder, yield: 92%, mp: 108.4–110.1 °C. [1]H NMR (500 MHz, DMSO-$d_6$): δ 7.86 (2H, *d*, *J* – 8.7 Hz, H-2; H-6); δ 7.03 (2H, *d*, *J* – 8.7 Hz, H-3; H-5); δ 7.73 (1H, *d*, *J* = 15.5, H-β); δ 7.80 (1H, *d*, *J* = 15.5, H-β); δ 7.57 (2H, *d*, *J* = 7.3 Hz, H-2′; H-6′); δ 8.14 (2H, *m*, H-3′; H-5′); δ 7.66 (1H, *d*, *J* = 7.3 Hz, H-4′); δ 3.83 (3H, *s*, *p*-OCH₃); [13]C NMR (125.8 MHz, DMSO-$d_6$): 55.8 (*p*-OCH₃); 114.8 (C-3; C-5); 120.1 (C-δ); 127.6 (C-1); 128.9 (C-3′; C-5′); 129.1 (C-2′; C-6′); 131.3 (C-2; C-6); 133.4 (C-4′); 138.2 (C-1); 144.5 (C-δ); 161.7 (C-4); 189.6 (C=O). ESI-MS: *m*/*z* 239.0 [M+H]$^+$. MS/MS [M+H]$^+$: 105.2 (100%), 133.1 (69.4%), 161.0 (74.2%). Molecular formula: $C_{16}H_{14}O_2$.

## 2.4. Determination of Purity Using [1]H qNMR

Using quantitative [1]H NMR (qNMR), the purity of the MPP was determined using the external standard dimethyl terephthalate (DMT) (certified purity 99.988 ± 0.060%, Inmetro, Rio de Janeiro, Brazil). This standard (*n* = 3) was prepared in DMSO-$d_6$ (530.0 μL), with TMS as the internal reference (0.0 ppm). Initially, the 90º pulse calibration was performed for all samples using the ZG pulse sequence. The longitudinal relaxation constant (T1) was estimated for the hydrogens at δ 8.08 (*s*, 4 H) using the inversion-recovery experiment (t1ir1d) (Figures S8–S10). The [1]H spectra were acquired using the ZGPR pulse sequence (residual water signal suppression), without sample rotation. The measured acquisitions of the standard (AQ) were 3.27 s, relaxation interval (d1) of 22 s (7 × T1 + AQ), 64k time domain points (TD) and a spectral window (SW) of 19.99 ppm.

MPP (4.80 ± 0.02 mg, *n* = 3) was dissolved in 530.0 μL of DMSO-$d_6$ containing TMS (≥99.0% purity) as the internal reference (0.0 ppm). These solutions remained in an ultrasound bath for 5 min (25.0 °C) and were then transferred to 5 mm-diameter NMR tubes [29]. The [1]H NMR acquisition parameters of MPP were standardized for the signal at 7.03 ppm (*d*, 2 H). The constant T1 was estimated by the inversion-recovery experiment (t1ir1d). The optimized acquisition parameters were as follows: P1 10.2 μs, AQ of 3.27 s, d1 of 15 s (7 × T1 + AQ), TD of 64 K, SW of 19.99 ppm, number of transients of 4, dummy scans of 2, receiver gain value 90.5. TopSpin software 4.1.3 was used for NMR spectra processing. The purity of the MPP was determined using Equation (1) [30]:

$$P_{MMP} = (I_{MPP}/I_{TD}) \times (N_{TD}/N_{MPP}) \times (M_{MPP}/M_{TD}) \times (W_{TD}/W_{MPP}) \times P_{TD} \qquad (1)$$

where I is the area of the absolute integral, (N) number of nuclei, (M) molar mass, ($W_{TD}$ and $W_{MPP}$) weight, ($P_{MPP}$) purity of the analyte and ($P_{TD}$) purity of the standard.

### 2.5. Analysis of 4-Methoxychalcone Using Headspace and Gas Chromatography Coupled to Mass Spectrometry (HS-GC-MS)

2.5.1. Extraction Procedure

Solutions containing MPP diluted with methanol were prepared using a range of concentrations (50 to 1000 mg/mL). The solutions were stored appropriately in vials under refrigeration until use. The analysis of the MPP was carried out using the headspace procedure [31] under optimized conditions and with a 1 mL syringe. Extraction was performed with a 10 mL vial sealed with a septa and screw cap, at 150 °C and agitation, for 10 min, with 100 μL of the reference or sample solution. After extraction, 1 mL of the vaporized sample inside the vial was aspirated, inserted into the injection port of the GC and analyzed as described in Section 2.5.2.

2.5.2. Chromatographic and Spectrometric Conditions

In the chromatographic and spectrometric analyses [31,32], diluted MPP (preparation described in Section 2.5.1) was analyzed using a gas chromatograph (2010 GC, Shimadzu, Kyoto, Japan) equipped with a mass spectrometer. The injection was carried out in splitless mode. The GC system was equipped with a DB-5 fused silica capillary column (30 m 0.25 mm–0.25 μm). The carrier gas was ultrapure helium (1.0 mL/min) and the linear velocity was 36.8 cm/s. The oven temperature was programmed as follows: 60 °C for 5 min, from 60 to 300 °C at 10 °C/min and 300 °C for 5 min. The injector and detector temperatures were set at 60 and 100 °C, respectively. In the mass spectrometry, the ion source temperature was 250 °C and the interface temperature was 280 °C, the detector was 0.7 kV and the mass/charge was 50 to 550 $m/z$. The signal was recorded and processed with LabSolutions GC-Solution software.

### 2.6. Molecular Docking

Molecular docking calculations were performed with the aid of the AutoDock Vina program (The Scripps Research Institute, San Diego, CA, USA) [33] in order to evaluate the interaction of MPP with bovine serum albumin (BSA), which was obtained from the Protein Data Bank (http://www.rcsb.org/pdb/, (accessed on 12 February 2023) under the code 4OR0 [34]. The docking protocol consisted of removing water molecules and the ligand (naproxen), assigning Gasteiger charges and delimiting grid boxes of dimensions 12 Å × 10 Å × 10 Å centered in the active site I (x = 69.671, y = 26.607, z = 86.784) and 14 Å × 16 Å × 14 Å centered in the active site II (x = 69.574, y = 28.575, z = 105.018). To validate the docking protocol, the ligand naproxen was removed and docked at sites I and II and then compared to the crystallized structures in each of the sites using the calculation of RMSD values. The obtained RMSD values were 1.93 and 0.85 for site I and II, respectively. Values up to 2 Å are considered reliable for a docking validation. The ligand geometry was optimized using DFT (Density Functional Theory) at B3LYP/6-31G (d) level. The docking results were visualized using Discovery Studio®® 4.5. (Waltham, MA, USA).

### 2.7. In Vitro Assays

2.7.1. Bovine Serum Albumin (BSA) Glycation Assay

The in vitro glycation assay was performed using two methods: model BSA/GO (bovine serum albumin and glyoxal) and model BSA/fructose, as previously described by [35] and adapted by [36]. For the BSA/GO model, the solutions of glyoxal (30 mM) and BSA (8 mg mL$^{-1}$) were prepared in phosphate buffer (200 mM, pH 7.4) containing sodium azide (3.0 mM, pH 7.4) as an antimicrobial agent. To determine the antiglycation activity, 30 μL of MPP (1 mg mL$^{-1}$ in DMSO), BSA (135 μL) and glyoxal (135 μL) were added to a microplate, and the initial fluorescence reading was performed (F1). After 24 h of incubation at 37 °C, the final fluorescence reading (F2) was performed. Quercetin was used as a positive standard (100.0 μg mL$^{-1}$) and DMSO as a negative control. The BSA/fructose model was performed as previously described, with the exception of glyoxal, which was replaced by fructose (100 mM). To determine antiglycation activity via this pathway, the

microplate was incubated for 120 h at 37 °C. Aminoguanidine was used as a positive standard (100.0 μg mL$^{-1}$). Fluorescence readings were performed on a microplate reader (excitation at 330 nm and emission at 420 nm) (DTX 800, Beckman Coulter, CA, USA). The 50% inhibitory concentration (IC$_{50}$) of MPP and the standard was determined via serial dilutions in DMSO (0.2–100.0 μg mL$^{-1}$). The experiments were carried out in triplicate. The inhibition percentage was calculated according to Equation (2):

$$[\% \text{ inhibition} = 100 - (F_{2\,(MMP/standard)} - F_{1\,(MMP/standard)}/F_{2\,(Negative\,control)} - F_{2\,(Negative\,control)}] \times 100 \quad (2)$$

### 2.7.2. In Vitro α-Glucosidase Inhibitory Assay

The inhibitory activity of the enzyme α-glucosidase was based on the measurement of the release of 4-nitrophenol from the 4-nitrophenyl-α-D-glucopyranoside (4-NPGP). The experiments were carried out using the α-glucosidase enzymes, which were extracted from rat intestinal acetone powders, and α-glucosidase from *S. cerevisiae*. The substrate 4-nitrophenyl-α-D-glucopyranoside (4-NPGP) and the two enzymes were diluted in phosphate buffer (10 mM, pH 6.9), separately. Each experiment was carried out independently, following the same experimental protocol [37]. In a microplate, 30 μL of MPP (1 mg mL$^{-1}$, in DMSO) and 170 μL of the enzyme (0.08 μg mL$^{-1}$) were incubated for 5 min at 37 °C under light protection, and the first reading was performed (R1). Subsequently, 100 μL of the 4-NPGP (5 mg mL$^{-1}$) was added and, after 20 min of incubation, the second reading was performed at 405 nm (R2). DMSO was used as a negative control and the acarbose standard was used as a positive control. The experiments were carried out in triplicate. The readings were performed on a microplate reader (DTX 800, Beckman Coulter, CA, USA). To calculate the inhibition percentage, Equation (3) was used:

$$[\% \text{ inhibition} = 100 - (R2_{sample/standard} - R1_{sample/standard}/R2_{negative\,control} - R1_{negative\,control}) \times 100] \quad (3)$$

### 2.7.3. Hemolytic Test

The hemolytic potential assay was performed according to [38]. Blood from a Swiss mouse was collected via the intracardiac route for preparation of the 2% erythrocyte suspension (ES). On a microplate, 100 μL of MPP (420 μM) and 100 μL of ES were added. Subsequently, the microplate was incubated for 1 h under constant stirring at 37 °C. After this period, the samples were centrifuged (1500 rpm/10 min) and the supernatant was transferred to another microplate, and the absorbance reading was performed on the spectrophotometer (DTX 800, Beckman Coulter, CA, USA) at 540 nm. As the positive control, Triton X-100 (0.5%) was used, and for the negative control, DMSO was used. The assay was performed in triplicate. The results were interpreted based on the % of hemolysis, where below 10% is considered non-hemolytic and above 25% is considered hemolytic activity of the compound [39].

### 2.7.4. Cell Viability Assay

The cytotoxicity of MPP in human lung fibroblast cells (MRC-5) was determined according to [40], with adaptations. The method is based on the enzymatic reduction of MTT (methylthiazolyldiphenyl-tetrazolium bromide) for formation of formazan crystals. The MRC-5 cells were plated (5 × 10$^3$ cells/well) and treated with 100 μL of MPP at different concentrations (0.6–20.0 μM). The microplate was incubated for 72 h (37 °C, 5% de CO$_2$). After this period, the wells were washed with PBS (pH 7.2) and then 100 μL of MTT (0.5 mg mL$^{-1}$) was added. The microplate was incubated for another 4 h at 37 °C, under protection from light. Viable cells metabolize MTT into purple-colored formazan crystals. Subsequently, 100 μL of DMSO was added to solubilize the formazan crystals and, after 5 min at room temperature, the reading was performed at 560 nm. Doxorubicin (20 μM) was used as a positive control and PBS as a negative control of cell death. The experiment was conducted in triplicate. The results were expressed as a percentage and the viability was calculated using Equation (4):

$$[\% \text{ viability} = (\text{Abs sample}/\text{Abs control}) \times 100] \quad (4)$$

*2.8. In Vivo Assays*

2.8.1. Animals

The experimental protocols followed the guidelines of the Brazilian National Council for Control of Animal Experimentation (CONCEA) and were approved by the Ethics Commission on the Use of Animals (CEUA) at the Federal University of Amazonas (UFAM) (No. 004/2019). For this study, Swiss albino mice (Unib:SW), males, 6 weeks of age, weighing 18–20 g, were used, which were obtained from the central vivarium at UFAM, Manaus, Brazil. The animals were kept under standard laboratory conditions (light-dark cycle of 12 h, temperature of $22 \pm 2$ °C and relative humidity of $50 \pm 10\%$) and fed with standardized rodent feed.

2.8.2. Oral Sucrose Tolerance Test (OSTT)

The oral sucrose tolerance test was performed as described by [41], with alterations. Healthy mice were divided into three groups of six animals. On the eve of the experiment, the animals were fasted for 12 h, and blood glucose levels were measured with the help of a portable precision glucometer (Xtra, Abbott Diabetes Care, Amadora, Portugal). One group was used as an untreated control and received 0.9% saline solution (NTG, Group 1). Another group received treatment with acarbose at a dose of 100 mg/kg body weight (bw) (acarbose100, Group 2). One test group was treated with MPP at a dose of 200 mg/kg bw (MPP200, Group 3). After 20 min, all the groups received a sucrose overload (2 g/kg bw), and blood samples were collected from the tail vein of the animal. The blood glucose level was measured at different times using the portable glucometer (30 min, 60 min, 90 min and 120 min). All administrations were performed orally (gavage).

2.8.3. Oral Glucose Tolerance Test (OGTT)

The oral glucose tolerance test was performed as described by [41], with alterations. The healthy mice were divided into four groups of six animals. One group was used as an untreated control and received 0.9% saline solution (NTG, Group 1). Another group received treatment with metformin at a dose of 200 mg/kg body weight (bw) (metformin200, Group 2). Two test groups were treated with MPP at doses of 100 mg/kg bw (MPP100, Group 3) and 200 mg/kg bw (MPP200, Group 4). All groups received a glucose overload (2 g/kg bw). The OGTT was performed as described in item Section 2.8.4

2.8.4. Oral Glucose Tolerance Test (OGTT) in Diabetic Mice

Type 2 DM induction was performed according to the method described by [42], with modifications. A group of six healthy animals were separated as a control group (NTG, Group 1). In the other mice, induction occurred via a single intraperitoneal injection of nicotinamide solubilized in saline solution (50 mg/kg bw). After 20 min, streptozotocin (STZ) was administered in a single dose of 150 mg/kg bw (1 M citrate buffer, pH 4.5). After 96 h, mice with blood glucose greater than 200 mg/dL were considered diabetic. The diabetic animals were randomized into the following five groups of six animals: untreated diabetic control group (DNTG, Group 2), standard group treated with metformin at a dose of 200 mg/kg bw (metformin200, Group 3) and two treatment groups with MPP at doses of 100 mg/kg (MPP100, Group 4) and 200 mg/kg (MPP200, Group 5). The experiment started with the measurement of the glucose level of all 12 h-fasting animals. Then, the treatment was carried out according to each group. After 20 min, all the groups received a glucose overload (2 g/kg bw). Blood glucose levels were measured at 30 min, 60 min, 90 min and 120 min. The metformin, MPP and glucose standard were solubilized in distilled water at the established doses and administered via gavage.

*2.9. Statistical Analysis*

The calculation of MPP purity, $IC_{50}$ and the analytical curve using a linear regression model was carried out using Excel®® 2013 and GraphPad Prism 6.0 (San Diego, CA, USA). The results of the in vitro and in vivo experiments were analyzed using the GraphPad Prism

6.0 program (San Diego, CA, USA) using ANOVA (one-way and two-way). The results were expressed as means (standard deviation). All *p*-values of < 0.05 were considered statistically significant.

## 3. Results

### 3.1. Chemical Analysis

The structural confirmation of chalcone (*E*)-3-(4-methoxyphenyl)-1-phenylprop-2-en-1-one (MPP) was performed based on the interpretation of [1]H NMR data (Figure 1), [13]C, DEPT135, HSQC, HMBC and ESI-MS/MS (Figures S1–S5), and conformed to those described in the literature [19]. To obtain the level of purity of MPP, absolute quantification by [1]H quantitative NMR was used, using the PULCON method [43]. Since it is a primary ratio method, it is not necessary to construct a calibration curve, nor employ standards that are identical to the analyte of interest [30,44]. In this study, dimethyl terephthalate (δ8.08, *s*, 4H) was used as an external standard to calibrate the NMR measurements, since the displacement is in the same chemical environment as the signal of interest of the analyte at δ7.03 (*d*, 2H). Based on the processed spectral data, the calculated average purity was 96.32% (RSD of 0.73%, *n* = 3), which indicates a high yield by the method employed (Table S1) [30].

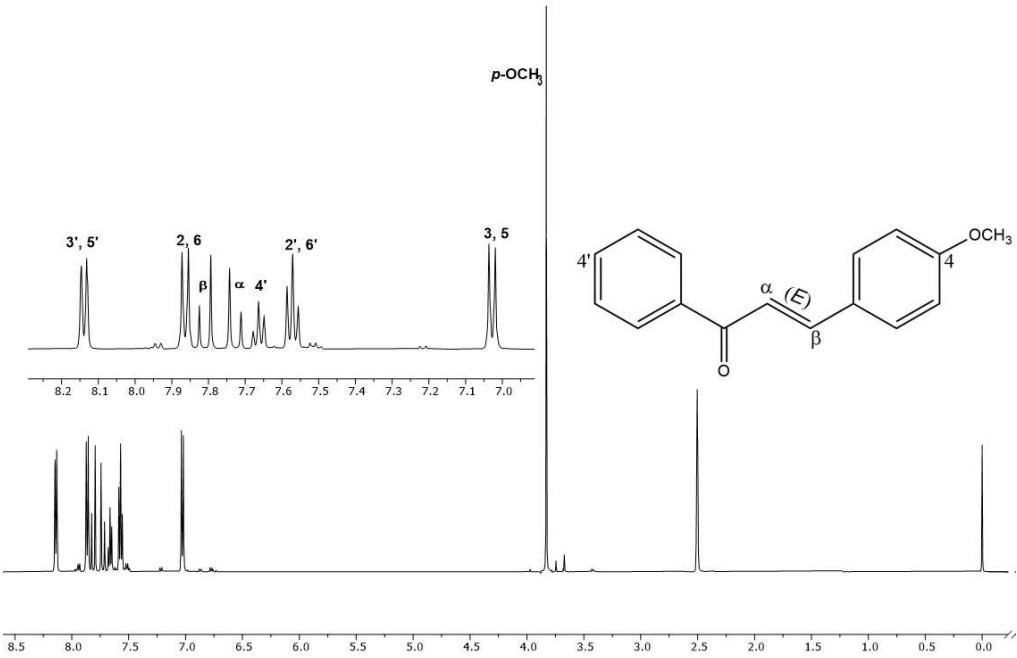

**Figure 1.** [1]H NMR spectrum and region amplification δ 7.0–8.3 of 4-methoxychalcone (MPP) (DMSO-$d_6$, 500 MHz).

### 3.2. Identification and Characterization of 4-Methoxychalcone via HS-GC-MS

The headspace technique was suitable for the extraction and transfer of 4-methoxychalcone from a liquid sample to a gaseous sample and analyzed by GC-MS. The 4-methoxychalcone peak was identified with the retention time at 40.833 min and the calibration curve used in the quantification of the substance where the linearity of the responses were acceptable ($R^2$ = 0.9998), and the molecule was confirmed with the mass spectrum showing the masses of the ions characteristic of MPP (Figure 2).

The work of [32] involved the study of the differentiation of isomeric methoxychalcones via electrospray ionization mass spectrometry and detected ions 133 *m/z*, 161 *m/z* and 239 *m/z*, which are ions that are compatible with the mass spectrum obtained from the 4-methoxychalcone in this study. Ions 77, 108, 133, 161, 223, 237, 238 and 239 were the majority ions detected in the MPP spectrum obtained in this work; the same majority ions were found in the MPP mass spectrum in searches on the site ChemicalBook [45].

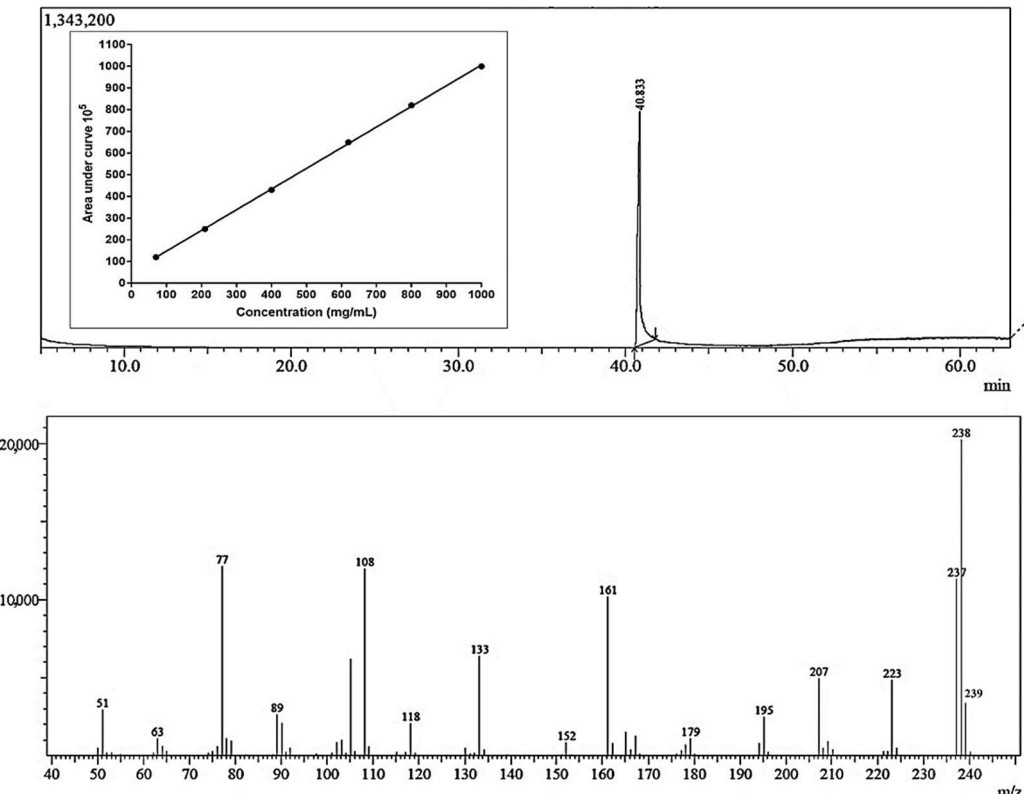

**Figure 2.** Chromatogram, calibration curve and mass spectrum of GC-MS headspace analysis for the quantification of 4-methoxychalcone. Each concentration was measured in quintuplicate following a fully randomized statistical design.

### 3.3. Glycation Inhibition Effect

The accumulation of advanced glycation end products (AGEs) is accelerated in hyperglycemic conditions, and AGEs have been associated with the various pathological disorders of diabetic complications [46]. Therefore, the literature reports methodologies for investigating the formation of AGEs and which analyze both synthetic and natural agents [36,46].

Since serum albumin plays a dominant role in bioavailability for drug transport, the efficiency of compounds as pharmaceutical agents depends on their ability to interact with this protein [47]. As there is no scientific information on the antiglycation effect in serum albumin models of MPP, the present study evaluated the antiglycation effect of MPP through oxidative (BSA/GO) and non-oxidative (BSA/fructose) pathways. The analysis of the results shows that MPP inhibits the glycation of BSA both via the oxidative pathway (IC$_{50}$ of $8.8 \pm 0.4$ µg mL$^{-1}$ or $37.4 \pm 0.8$ µM) and via the non-oxidative pathway (IC$_{50}$ = $16.2 \pm 0.4$ µg mL$^{-1}$ or $65.6 \pm 1.7$ µM). MPP showed a significant response in relation to the positive standards quercetin (IC$_{50}$ = $35.6 \pm 0.9$ µg mL$^{-1}$ or $117.8 \pm 3$ µM) and aminoguanidine (IC$_{50}$ = $28.6 \pm 1.4$ µg mL$^{-1}$ or $258.6 \pm 12.7$ µM, respectively ($p < 0.05$) (Figure S6). Therefore, these data suggest that MPP can be considered a potent antiglycation agent that can act in both the early and advanced stages of glycation in diabetic patients. This statement is justified when comparing these results with other research conducted with chalcone derivatives for antiglycation, such as the study conducted with oxindole-based chalcones and their activity against protein glycation (showing an IC$_{50}$ = $155.22 \pm 2.98$ µM to IC$_{50}$ = $289.47 \pm 2.47$ µM), presenting inhibitory activity against glycation compared to the reference standard rutin (IC$_{50}$ = $294.5 \pm 1.5$ µM) [48]. This demonstrates that MPP showed superior antiglycation activity and reinforces the potential that MPP and chalcones present as antiglycants agents.

### 3.4. Molecular Docking for BSA

Understanding the structural aspects of ligand binding by albumins is important for drug delivery due to the fact that the protein enables drug solubilization, thus allowing transport by blood. Serum albumin (SA) shows a variety of binding sites; however, two major regions in albumin are responsible for reversible binding of many drugs [49,50]. Site 1, also named Drug Site 1, is a binding pocket located in the core of subdomain IIA and comprises all six helices of the subdomain residues 148–154 of subdomain IB. The interior of the pocket is predominantly non-polar, though it contains two regions of polar residues [50]. This site binds bulky heterocyclic compounds without a carboxyl group, for example, warfarin or phenylbutazone. Site 2, also named Drug Site 2, is composed of all six helices of subdomain IIIA and is similar to site 1 (subdomain IIA), though it is smaller. This site is best allocated by compounds that have an aromatic ring and carboxylate groups that show extended conformation, e.g., ibuprofen and suprofen [50]. It is noteworthy that both BSA and human serum albumin (HSA) are homologous in structure and, due to the low acquisition cost of BSA, it is common to substitute HSA for BSA to represent serum albumins in experimental albumin–ligand interaction studies [51,52]. In this context, docking calculations of the molecule under study were performed with BSA, focusing on interactions at Drug Sites 1 and 2 (Figure 3), with the aim of supporting the obtained experimental data.

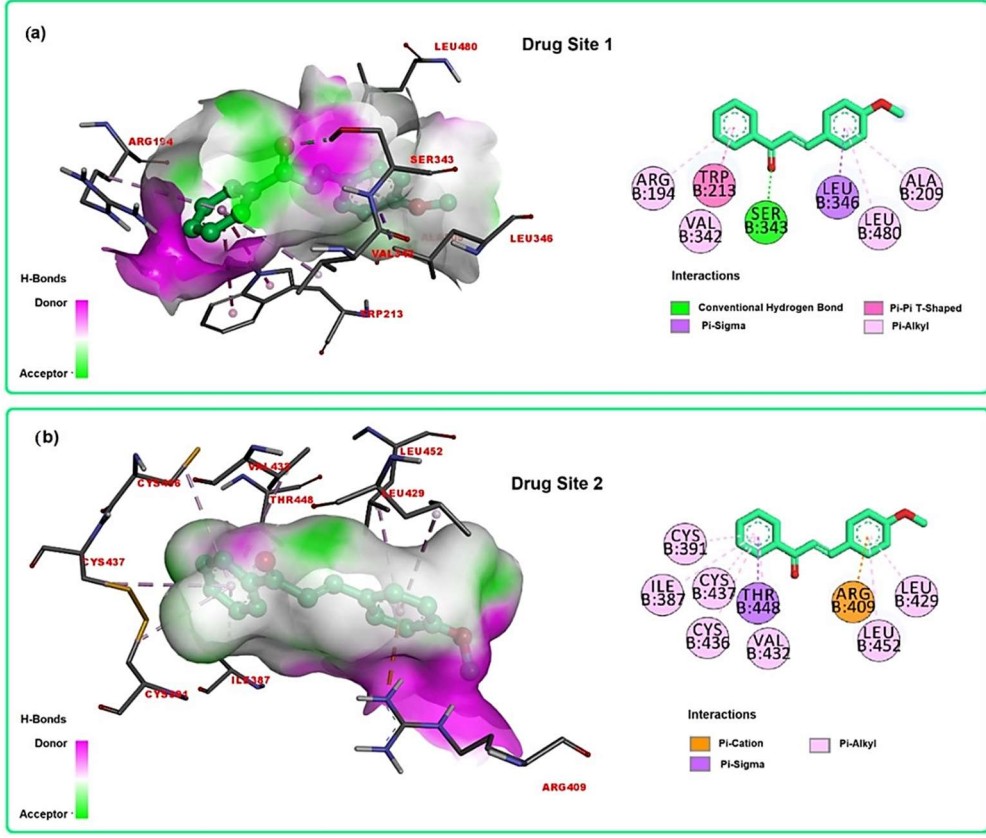

**Figure 3.** Docking calculations of MPP in the BSA active sites: (**a**) 3D and 2D representations of interactions at Drug Site 1; (**b**) 3D and 2D representations of interactions at Drug Site 2.

For Drug Site 1, MPP showed binding energy equal to −8.5 kcal/mol for the higher score conformation, while naproxen, the co-crystallized ligand, presented binding energy equal to −9.0 kcal/mol. Binding mode analysis showed that MPP complexed to Drug Site 1 (Figure 3a) by H-bond interaction with Ser 343, π-sigma with Leu 346, π-π (T-Shaped) with Trp 213 and π-alkyl interactions with Arg 194, Val 342, Leu 480 and Ala 209. For Drug Site 2, MPP showed binding energy equal to −7.5 kcal/mol for the higher score conformation,

while naproxen presented binding energy equal to −8.9 kcal/mol. Binding mode analysis revealed that MPP complexed to Drug Site 2 (Figure 3b) by π-cation interaction with Arg 409, π-sigma with Thr 448 and π-alkyl interactions with Arg 194, Val 342, Leu 480 and Ala 209. Although the molecule showed negative binding values for both sites, Drug Site 1 seems to better accommodate the molecule under study, as the docking simulations revealed a more negative value for MPP at this site (−8.5 kcal/mol). Furthermore, despite the interactions being predominantly non-polar in both sites, in Drug Site 1, a strong H-bond is formed, followed by π-π (T-Shaped) interaction, indicating more stability for the BSA-MPP complex when MPP binds to Drug Site 1. Moreover, the size of Drug Site 1 contributes to better allocation of the molecule, as this site is larger than Drug Site 2 and thus favors better interactions. The binding mechanism of chalcone to bovine serum albumin (BSA) has been widely explored by computational and spectroscopic methods. This protein affinity is directly linked to the distribution and half-life of chalcones [52,53].

To predict the pharmacokinetics of MPP, the web tool SwissADME was used (http://www.swissadme.ch, accessed on 12 February 2023). MPP demonstrated compliance with Lipinski's Rule of Five and Veber's Rule, which enables its oral use [54,55]. Additionally, the compound showed a moderate-to-high tendency to dissolve in organic solvents, with moderate solubility in water (Table S2). These properties suggest the potential of MPP to cross the blood–brain barrier, but they may also influence its bioavailability, estimated at 55%, due to gastrointestinal absorption and protein binding (Figure 3). However, the compound's lipophilicity may reduce its oral absorption. To overcome this challenge, future studies will seek to develop a nanoparticle formulation, aiming to enhance its effects without increasing the risk of toxicity.

### 3.5. Hemolytic Activity and Cytotoxic Effect in MRC-5 Cells

The cytotoxic effect of MPP was evaluated in human lung fibroblast cells (MRC-5) at the concentrations of 0.6 μM to 20.0 μM during 24 h, 48 h and 72 h of incubation (Figure 4A). MPP presented cell viability of 108.4 ± 8.1% (24 h), 110.3 ± 9.1% (48 h) and 85.5 ± 1.5% (72 h) at a concentration of 20 μM. Therefore, an $IC_{50} > 20.0$ μM means that the concentration that causes 50% of cell death is above this value. The positive control doxorubicin (20 μM) significantly reduced cell viability by approximately 90.0% after 72 h. However, compared to the control group of untreated cells, MPP showed no significant difference in either the tested concentrations and in the different incubation periods ($p > 0.05$).

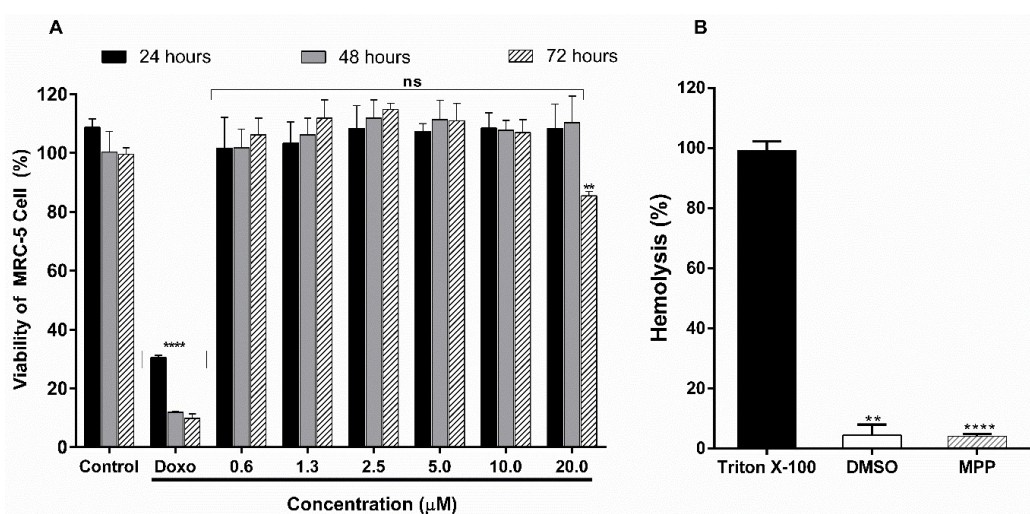

**Figure 4.** (**A**) Cell viability of the untreated control group, positive control doxorubicin at 20 μM (doxo) and MPP at different concentrations per period of 24, 48 and 72 h in MRC-5 cell line; (**B**) hemolytic activity of MPP at the concentration of 20 μM compared to the standard triton X-100. Results are shown as the mean ± standard deviation ($n = 3$). **** $p < 0.0001$. ns = not significant ($p > 0.05$) vs. untreated control group; ** $p < 0.01$ (ANOVA followed by Dunnett's multiple comparisons test).

In addition, as shown in Figure 4B, MPP showed low hemolytic activity (4.1 ± 0.6%) (20 μM) compared to the triton X-100 standard (99.1 ± 2.9%). This result is below 10%, which is interpreted as non-hemolytic [39]. In comparison with the negative control (DMSO), no significant difference was found ($p > 0.05$). Therefore, these results reveal to us that MPP does not present hemolytic toxicity in the concentrations and experimental models tested.

### 3.6. α-Glucosidase Inhibition

Inhibition of the catalytic activity of the enzyme α-glucosidase leads to a delay in glucose absorption and a reduction in postprandial glycemia [56]. In view of the anti-diabetic potential of chalcone derivatives [22,57]. MPP was initially evaluated for the inhibitory potential of the enzyme α-glucosidase extracted from *Saccharomyces cerevisiae* and rat intestinal acetone powders. This constituent exhibited high inhibitory activity of the enzyme from *S. cerevisiae* (93.2 ± 1.5%), with an IC$_{50}$ of 24.5 ± 0.8 μg mL$^{-1}$, when compared to standard acarbose (24.1 ± 1.5%). This result differs from a previously described study, in which the inactivity of this chalcone against α-glucosidase from yeast was demonstrated [56]. The positive control quercetin showed inhibition of 99.6 ± 0.3% (Figure 5) with an IC$_{50}$ of 5.9 ± 0.2 μg mL$^{-1}$ ($p < 0.05$), corroborating the suggestion that this flavonoid is a competitive-type inhibitor and that the positions at C = O and 3-OH (C ring) and the conjugations of the B ring play an important role in the inhibition of the enzyme [58].

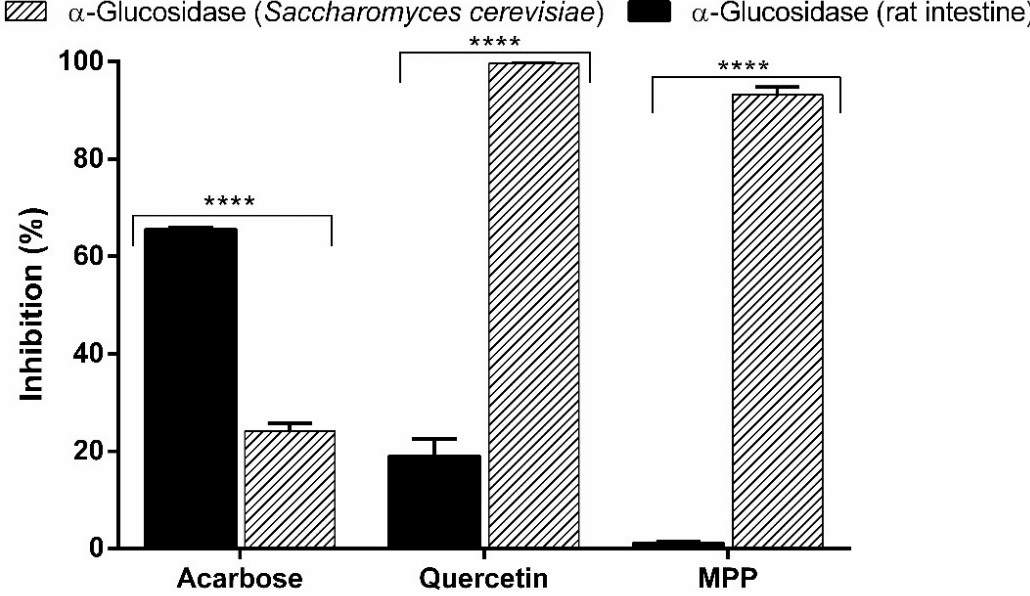

**Figure 5.** Inhibitory activity of acarbose, quercetin and MPP at the concentration 100 μg mL$^{-1}$ on the enzymes of α-glucosidase extracted from *Saccharomyces cerevisiae* and α -glucosidase extracted from rat intestinal acetone powders. The results are expressed as mean ± SD, $n = 3$. **** $p < 0.0001$ (ANOVA followed by Dunnett's multiple comparisons test).

Regarding the inhibitory evaluation in α-glucosidase from rat intestine, MPP was inactive (1.1 ± 0.2%) compared to standard acarbose (65.5 ± 0.4%) ($p < 0.05$). Our data prove that, with different enzymes, the in vitro model used can compromise a reliable comparison between studies, which led us to investigate the mechanism of action of MPP in an in vivo experimental model.

### 3.7. Anti-Hyperglycemic Effect

MPP was evaluated in the oral sucrose tolerance test (OSTT) at a concentration of 200 mg/kg bw. Initially, healthy mice were subjected to a sucrose overload of 2 g/kg bw,

causing high blood glucose levels over 200 mg/dL during the 120 min experiment (NTG) (Figure 6A). Acarbose was used as a positive control and administered at a dose of 100 mg/kg bw (Acarbose100). In this group, a reduction in the glucose level of 47.1% was observed over 90 min and was significantly different from the NTG Group ($p < 0.0001$). In contrast, MPP at a dose of 200 mg/kg bw (MPP200) did not reduce glucose levels and there was no significant difference in relation to this group ($p > 0.05$). Therefore, these results corroborate the enzymatic inhibition study carried out with rat intestine α-glucosidase in vitro (previously described) and suggest that MPP does not inhibit the intestinal α-glucosidase enzyme.

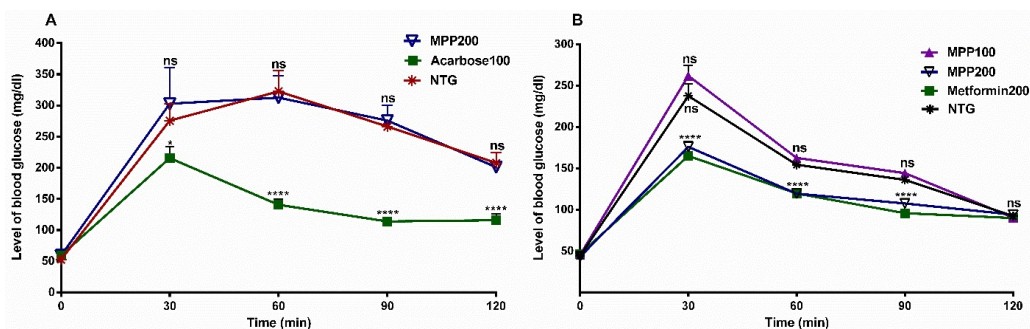

**Figure 6.** (**A**) Effects of the oral administration of MPP and acarbose on blood glucose concentration in sucrose-loaded mice. Values are expressed as mean ± SD, *n* = 6. * *p* = 0.03; **** *p* < 0.0001 compared to untreated control group. Acarbose100: acarbose 100 mg/kg bw (positive control); MPP200: 4-methoxychalcone (200 mg/kg bw). (**B**) Effects of the oral administration of MPP and metformin on blood glucose concentration in glucose-loaded mice. Metformin200: metformin 200 mg/kg bw (positive control); MPP100 and MPP200: 4-methoxychalcone (100 mg/kg and 200 mg/kg bw, respectively). NTG: untreated healthy mice control group. Values are expressed as mean ± SD, *n* = 6. ns = not significant (*p* > 0.05), **** *p* < 0.0001 vs. untreated control group (ANOVA followed by Dunnett's multiple comparisons test).

Given this result, MPP was submitted to the oral glucose tolerance test (OGTT) in healthy mice. In this experiment, the animals received a glucose overload of 2 g/kg bw, with an induced increase occurring in glucose levels (DNTG), which remained elevated at over 140 mg/dL for 90 min of the experiment (Figure 6B). In the hyperglycemic animals treated with MPP at a dosage of 200 mg/kg bw (MPP200), significant reductions in glucose levels were observed after 30 min ($p < 0.0001$). Similarly, blood glucose levels were continuously reduced to 61.1% and 53.4% (90 and 120 min, respectively), after 60 min; whereas the group treated with metformin at a dose of 200 mg/kg bw (Metformin200) caused a reduction to 58.0 and 54.4% (90 and 120 min, respectively), and there was no significant difference in relation to the group treated with MPP200 ($p > 0.05$). For the group of animals treated with MPP at a dosage of 100 mg/kg bw (MPP100), compared to NTG, the reduction in glycemic levels seen with MPP200 did not show a significant difference compared to the group of animals treated with metformin at the same dosage ($p > 0.05$). These data suggest that MPP200 significantly inhibits the increase in blood glucose.

To evaluate the hypoglycemic potential of MPP, animals with streptozotocin-induced (STZ-induced) type 2 diabetes were treated with MPP at dosages of 100 and 200 mg/kg bw (MPP100 and MPP200, respectively). After 12 h of fasting, these animals received a glucose overload of 2 g/kg bw and blood glucose levels rose above 300 mg/dL (T0 min) (Figure 7).

Administration of MPP 200 mg/kg bw significantly reduced the state of hyperglycemia in STZ-induced diabetic animals after 30 min. This reduction in glycemic levels was statistically similar to animals treated with metformin at 200 mg/kg bw ($p > 0.05$). However, there was no significant difference in glucose levels in animals treated with MPP100 compared to the group of untreated diabetic animals (DNTG) after 60 min ($p < 0.05$).

Therefore, our study revealed the hypoglycemic potential of MPP (MPP200) through the mechanism of cellular glucose uptake. These data corroborate a previous study that reports the in vitro potential of MPP in glucose reduction by this or another mechanism such as insulin receptor activation and interaction with related-protein glucose control [22,57,58]. Furthermore, studies related to the performance of the microbiome in drug metabolism, which directly influences its absorption and bioavailability, along with the potential of flavonoids to improve glycemia, as is the case with MPP, direct this research towards future studies on the possible mechanisms of action of MPP on the microbiota [59]. Studies with chalcone derivatives have also demonstrated this performance in the microbiota [60,61]. Investigations with derivatives of chalcones have related the anti-diabetic activity to its binding potential to the peroxisome proliferator-activated gamma receptor (PPAR-γ) [22,62]. Thus, it is possible to assume that MPP can be used as an anti-diabetic agent, and additional preclinical analyses are already underway.

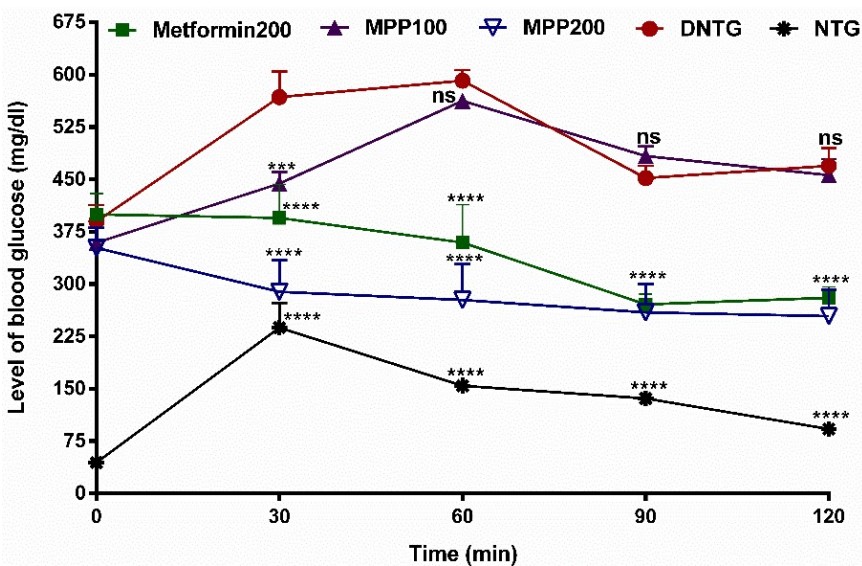

**Figure 7.** Effects of the oral administration of MPP and metformin on the blood glucose concentration in glucose-loaded diabetic mice. Metformin200: metformin 200 mg/kg bw (positive control); MPP100 and MPP200: 4-methoxychalcone (100 mg/kg and 200 mg/kg bw, respectively). DNTG: untreated diabetic control group. NTG: untreated healthy mice group. Values are expressed as mean ± SD, $n = 6$. ns = not significant ($p > 0.05$). The character **** indicates $p < 0.0001$ and *** indicates $p < 0.001$ versus the untreated control group (ANOVA followed by Dunnett's multiple comparison test).

## 4. Conclusions

A method by HS-GC-MS was developed to quantify 4-methoxychalcone in solutions with satisfactory precision and reproducibility for its characterization and quantification. In the docking assays, 4-methoxychalcone was shown to be well-accepted orally, but its bioavailability is hindered by its high liposolubility, which reduces its absorption. In terms of distribution, it is easily transported due to its affinity with serum albumin by binding to Site I through H-bonding, followed by π-π interaction. 4-methoxychalcone showed no cytotoxicity in the human lung fibroblast lineage (MRC-5) after 72 h, and in the hemolysis test, mouse erythrocytes maintained stable and integral membranes. Additionally, it was demonstrated that this chalcone has hypoglycemic activity in diabetic mice, and this effect may be related to cellular glucose uptake, the mechanism of which will be presented in future studies. Therefore, this study presents results that show that 4-methoxychalcone has the potential to be an anti-diabetic agent capable of reducing glucose levels to near normal levels. Further results and preclinical studies are underway to investigate the mechanisms involved in this effect.

**Supplementary Materials:** The following supporting information can be downloaded at: https://www.mdpi.com/article/10.3390/appliedchem4020010/s1. Figure S1: $^{13}$C spectrum of MPP (125 MHz, DMSO-d6), Figure S2: DEPT-135 spectrum of MPP (125 MHz, DMSO-d6), Figure S3: $^{1}$H–$^{13}$C HSQC of MPP (11.74 T, DMSO-d6), Figure S4: $^{1}$H–$^{13}$C HMBC of MPP (11.74 T, DMSO-d6), Figure S5: ESI-MS/MS spectra in positive mode at *m/z* 239, Figure S6: Antiglycant activity of MPP (A) and quercetin (B) via the BSA/glyoxal method. Antiglycant activity of MPP (C) and aminoguanidine (D) using the BSA/fructose method, Figure S7: The 90° pulse experiment (zg) calibrated for the signal at δ 8.08 (*s*, 4H) of dimethyl terephthalate standard, Figure S8: The inversion-recovery experiment (t1ir1d) for the signal at δ 8.08 (s, 4H) of the dimethyl terephthalate standard (d1 = 22s). Figure S9: The 90° pulse experiment (zg) calibrated for the signal at δ 7.03 (*d*, 2H) of MPP (360° pulse), Figure S10: The inversion-recovery experiment (t1ir1d) for the signal at δ 7.03 (*s*, 2H) of MPP (d1 = 15s), Table S1: MPP purity obtained via $^{1}$H qNMR and with the values of the absolute integrals of the TD and MPP standard. Table S2: Pharmacokinetic parameters of MPP.

**Author Contributions:** L.D.R.A.: Conceptualization, methodology, validation, formal analysis, investigation, writing—original draft. E.S.C.O.: methodology, formal analysis, investigation, writing—original draft. S.B.C.: GC-MS analyses and interpretation, methodology, formal analysis, validation, investigation, writing—original draft. F.P.A.M. and L.d.S.M.: methodology, formal analysis, investigation. R.A.C.: conceptualization, methodology, formal analysis, investigation, writing—original draft. R.S.B.: methodology, resources. H.H.F.K.: LC-MS/MS analyses and interpretation. M.B.M.: NMR analyses and interpretation. I.R.d.S.M.: methodology, formal analysis, investigation. E.S.L.: conceptualization, supervision, methodology, validation, resources, writing—review and editing. All authors have read and agreed to the published version of the manuscript.

**Funding:** This research was funded by the Coordenação de Aperfeiçoamento de Pessoal de Nível Superior—CAPES, Conselho Nacional de Desenvolvimento Científico e Tecnológico-CNPq (FAP/CNPq N.º 003/2022), Financiadora de Estudos de Projetos e Programas—FINEP, and Fundação de Amparo à Pesquisa do Estado do Amazonas—FAPEAM (programa POSGRAD e FAP/CNPq N.º 003/2022).

**Institutional Review Board Statement:** The study was conducted in accordance with the guidelines of the National Council for the Control of Animal Experimentation (CONCEA) of Brazil and approved by the Ethics Committee on Animal Use (CEUA) of the Federal University of Amazonas (UFAM) (protocol n° 004/2019).

**Informed Consent Statement:** Not applicable.

**Data Availability Statement:** The data presented in this study are available in the article and Supplementary Material.

**Acknowledgments:** The authors would like to thank the Coordenação de Aperfeiçoamento de Pessoal de Nível Superior—CAPES, Conselho Nacional de Desenvolvimento Científico e Tecnológico-CNPq, Financiadora de Estudos de Projetos e Programas—FINEP, Fundação de Amparo à Pesquisa do Estado do Amazonas—FAPEAM and Central Analítica at UFAM for financial support and infrastructure.

**Conflicts of Interest:** The authors declare no conflicts of interest.

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
