# Peer review of "Antidiabetic Activities and GC-MS Analysis of 4-Methoxychalcone"

_appliedchem, doi:10.3390/appliedchem4020010_

Round 1
Reviewer 1 Report
Comments and Suggestions for Authors
The manuscript submitted by Acho et al., is an interesting characterization study also testing the anti-diabetic potential of the assessed compound through in vitro and in vivo experiments.
The reviewer would like to propose the following points for the authors' consideration:
1. How was the number of animals determined (power calculation etc)?
2. In the discussion section it would be interesting to include the potential of the microbiome modification due to treatment. It is well established that the microbiome does play a role in glycemic and potentially diabetes, while flavonoids are know to improve glycemia potentially. Such discussion would be strengthening the paper if took place in the discussion section. A potentially useful paper in that regard is the following:
Lacombe A, Li RW, Klimis-Zacas D, Kristo AS, Tadepalli S, Krauss E, Young R, Wu VC. Lowbush wild blueberries have the potential to modify gut microbiota and xenobiotic metabolism in the rat colon. PLoS One. 2013 Jun 28;8(6):e67497. doi: 10.1371/journal.pone.0067497. PMID: 23840722; PMCID: PMC3696070.
Comments on the Quality of English Language
The English language is OK although the manuscript would benefit from a read through by a native English speaker.
Author Response
- How was the number of animals determined (power calculation etc)?
The number of animals was determined following the protocol that is statistically acceptable by the Animal Ethics Committee (CEUA), with 6 animals per group. The number of animals was chosen based on a variety of factors, including statistical, ethical, and practical considerations. This was done to ensure meaningful results while minimizing unnecessary animal use. The use of six animals per group was determined based on statistical considerations to ensure that the study has statistical power.
Below are some articles that we based our choice of the number of animals on:
- PMID: 24894401 PMCID: PMC4160094 DOI: 10.1167/iovs.14-14383 (Diabetic and non-diabetic mice of 5 to 7 animals)
- PMID:23255670, DOI: 4149/gpb_2012_039 (Diabetic rats n = 6)
- PMID: 23238787 PMCID: PMC3563947 DOI: 10.1007/s00125-012-2792-x (Diabetic rats n = 6)
- PMID: 33225469 DOI: 10.1096/fj.202001138RR (Mice n=6)
2.In the discussion section it would be interesting to include the potential of the microbiome modification due to treatment. It is well established that the microbiome does play a role in glycemic and potentially diabetes, while flavonoids are know to improve glycemia potentially. Such discussion would be strengthening the paper if took place in the discussion section. A potentially useful paper in that regard is the following:
Lacombe A, Li RW, Klimis-Zacas D, Kristo AS, Tadepalli S, Krauss E, Young R, Wu VC. Lowbush wild blueberries have the potential to modify gut microbiota and xenobiotic metabolism in the rat colon. PLoS One. 2013 Jun 28;8(6):e67497. doi: 10.1371/journal.pone.0067497. PMID: 23840722; PMCID: PMC3696070.
The recommended article was inserted, and a paragraph about the potential modification of the microbiome due to treatment was included. The potential for microbiota alteration is very interesting and will help complement the article for a possible mechanism of action of MPP and future studies on the subject.
Reviewer 2 Report
Comments and Suggestions for Authors
Current report evaluated anti-diabetic activities and analyze 4-Methoxychalcone (MPP) by the gas chromatography coupled to mass spectrometry (HS-GC-MS). I like to give the following comments.
1. Chalcones belong to the class of flavonoids and merit for diabetic disorders shall be introduced in detail.
2. In line 74, “-glucosidase” seems an error.
3. MPP can be considered a potent antiglycation agent (line 363) that needs to compare with others.
4. Docking for serum albumin needs a positive reference.
5. Figure 7 seems better to compare using AUC in each. Additionally, glucose-loaded diabetic mice must show the dose of glucose in clear.
6. STZ-induced diabetic animals need the clear data to support.
7. In line 509, “This reduction in glycemic levels was not significantly different from the group of animals treated with metformin at the same dosage (p>0.05).” It needs to rephrase this sentence to be clear showing that reduction of hyperglycemia by MPP200 is same as metformin at same dosing.
8. The antiglycation activity of MPP needs direct evidence in addition.
9. It seems not reliable to conclude that 4-methoxychalcone has the potential to be an anti-diabetic agent. Limitation(s) of current report was ignored.
10. The hemolysis in the erythrocytes seems unclear. More data were required.
Comments on the Quality of English Language
It seems better to check through professional editing.
Author Response
- In line 74, “-glucosidase” seems an error.
Thank you for checking. The error has been corrected.
- MPP can be considered a potent antiglycation agent (line 363) that needs to compare with others.
It was added a paragraph comparing these results with another study that investigated chalcone analogs against protein glycation.
Reference: Khan, A.; Khan, A.; Farooq, U.; Taha, M.; Shah, A. A. S.; Halim, A. S.; Akram, A.; Khan, Z. M.; Jan, K. A.; Al-Harrasi, A. Oxindole-based chalcones: synthesis and their activity against glycation of proteins. Med Chem Res, 2019, 28, 900–906. DOI: https://doi.org/10.1007/s00044-019-02345-1.
- Docking for serum albumin needs a positive reference.
Positive references were added regarding anchoring to serum albumin.
References: Singh, N.; Kumar, N.; Rathee, G.; Sood, D.; Singh, A.; Tomar, V.; Dass, S.K.; Chandra, R. Privileged Scaffold Chalcone: Synthesis, Characterization and Its Mechanistic Interaction Studies with BSA Employing Spectroscopic and Chemoinformatics Approaches. ACS Omega. 2020, 27,5,2267-2279. DOI: 10.1021/acsomega.9b03479
Karthikeyan, S.; Thirunarayanan, A.; Shano, L.B.; Hemamalini, A.; Sundaramoorthy, A.; Mangaiyarkarasi, R.; Abu, N.; Ganesan, S.; Chinnathambi, S.; Pandian, G.N. Chalcone derivatives' interaction with human serum albumin and cyclooxygenase-2. RSC Adv. 2024,17,14,2835-2849. doi: 10.1039/d3ra07438b.
- Figure 7 seems better to compare using AUC in each. Additionally, glucose-loaded diabetic mice must show the dose of glucose in clear.
We agree that the area under the curve is a more sensitive method, but the way the result was presented already shows statistically significant differences between the groups.
- STZ-induced diabetic animals need the clear data to support.
The use of STZ with nicotinamide for the induction of diabetes mellitus in rats and mice is the most commonly used chemical method. IDPM: PMC8297477 DOI: 10.2147/DDDT.S316185.
- In line 509, “This reduction in glycemic levels was not significantly different from the group of animals treated with metformin at the same dosage (p>0.05).” It needs to rephrase this sentence to be clear showing that reduction of hyperglycemia by MPP200 is same as metformin at same dosing.
The sentence was reformulated: The reduction in glycemic levels seen with MPP200 did not show a significant difference compared to the group of animals treated with metformin at the same dosage (p>0.05).
- The antiglycation activity of MPP needs direct evidence in addition.
The in vitro antiglycation assay was performed, and the HbA1c of the mice was measured. AGEs were not measured due to a lack of access to kits at the time of collection because of the pandemic.
- It seems not reliable to conclude that 4-methoxychalcone has the potential to be an anti-diabetic agent. Limitation(s) of current report was ignored.
Preclinical tests show that MPP at a dose of 200 mg/kg reduces blood glucose levels in diabetic mice, demonstrating its potential for diabetes treatment. The conclusion was reformulated, emphasizing more strongly the potential of 4-methoxychalcone as an antidiabetic agent.
- The hemolysis in the erythrocytes seems unclear. More data were required.
Red blood cells are susceptible to lysis, and the hemolysis test shows that MPP does not destroy red blood cells, confirming its safety. The reference used for the hemolysis test protocol was: Amin, K.; Dannenfelser, R.M. In vitro hemolysis: guidance for the pharmaceutical scientist. J Pharm Sci. 2006, 95,6,1173-6.
Reviewer 3 Report
Comments and Suggestions for Authors
In the manuscript L. Acho et coworkers tried to reveal the potential of 4-methoxychalcone as antidiabetic agent. At first sight the manuscript is potentially interesting for the readers, since it presents the results matching chemical synthesis with the basic biological activity studied in vitro and in vivo. But the second analysis reveals some doubts of the novelty and scientific significance of the submitted manuscript, since the 4-methoxychalcone can be bought without any difficulties, which means that it is synthesized already. So why the Authors presented the chemical synthesis of this compound and why the 4-methoxychalcone has been chosen?
The Authors decided to present the biological activity of this compound as the potential anti-diabetic agent. Why this particular area has been chosen?
Next doubts are as follow:
- Why there are presented results with different units of concentration? This obligatory needs to be unified and since it is one chemically pure compound present the results in M units. In case of in vivo studies correlate the doses with the M unit.
- Why for cytotoxic potential cell line not connected with diabetes/metabolism/administration was chosen? The origin is not connected with in vivo assay. Why the cytotoxicity was determined after incubation for 72 hours?
- In ace of other procedures some concentrations are presented in brackets, i.e. line 216. What it means – is this the final concentration in the mixture or 170 microL of enzyme solution at concentration 0.08 microg/L was taken? All methods should be improved and present clearly the experimental details.
- Why Swiss albino mice were used during in vivo experiments? If n=6 (number of animals in each group) is statistically enough to give the proper answer about the biological impact of MPP?
- Why during in vivo experiments doses equal to 100 and 200 mg/kg/bw were used? Also acarbose?
- Why in some experiments quercetin was used as an additional control and what is the rationale for choosing presented concentration? The same doubt is for usage of doxorubicine (it is not related to the main topic of the study). Why metformin or acarbose were not chosen?
- Why the effect on glycation was not studied with the samples collected from in vivo experiments?
- Why the effect of MMP on differently originated a-glucosidase was not studied more deeply with molecular docking? The mechanism of enzyme inhibition has not been presented (despite the fact that the Authors have the tools).Why human originated a-glucosidase was not studied?
- Discussion of the results is VERY simple! It should present more detailed information!!! The bioavailability issue should be presented since it will have a great impact even on the binding with serum albumin.
- The potential mechanism matching the effect of MPP on glucose uptake should be explained with more details. From my point of view, the residual attempts to discuss the obtained results with the literature data do not correspond to the level of the reviewed article.
- In my opinion presented results do not justify the usage in title of the word “anti-diabetic”, since the results do not correlate with the antidiabetic potential.
The explanations for the questions should be included in the improved version of the manuscript, as well as some experimental data.
In my opinion submitted manuscript requires at the least the major revision.
Author Response
- In the manuscript L. Acho et coworkers tried to reveal the potential of 4-methoxychalcone as antidiabetic agent.At first sight the manuscript is potentially interesting for the readers, since it presents the results matching chemical synthesis with the basic biological activity studied in vitro and in vivo. But the second analysis reveals some doubts of the novelty and scientific significance of the submitted manuscript, since the 4-methoxychalcone can be bought without any difficulties, which means that it is synthesized already. So why the Authors presented the chemical synthesis of this compound and why the 4-methoxychalcone has been chosen?
MPP is a low-cost, easily synthesized chalcone with a high level of purity. While the chemical synthesis of MPP was presented in the work, it was not the focus of the study. The highlights of this work are the studies using HS-GC-MS and the antidiabetic activity of MPP. This study is part of a larger research project, and based on these studies, there is the development of a nanoformulation aimed to increment its bioactivity
- The Authors decided to present the biological activity of this compound as the potential anti-diabetic agent. Why this particular area has been chosen?
Our research group specializes in developing potential treatments for chronic diseases such as diabetes, which have a major impact on public health.
Next doubts are as follow:
- Why there are presented results with different units of concentration? This obligatory needs to be unified and since it is one chemically pure compound present the results in M units. In case of in vivo studies correlate the doses with the M unit.
The units of "µg/ml" have already been converted to "µM," and the units of "mg/kg body weight" are used to provide personalized treatment for each animal and facilitate comparison with other in vivo studies.
- Why for cytotoxic potential cell line not connected with diabetes/metabolism/administration was chosen? The origin is not connected with in vivo assay. Why the cytotoxicity was determined after incubation for 72 hours?
The cytotoxicity test is based on the cell's reducing capacity to form a fluorescent compound. The amount of fluorescence emitted is directly proportional to the amount of living or viable cells. The incubation time according to the protocol is 12, 24, and 72 hours, which is sufficient time for the molecule under study to cause cellular damage.
- In ace of other procedures some concentrations are presented in brackets, i.e. line 216. What it means – is this the final concentration in the mixture or 170 microL of enzyme solution at concentration 0.08 microg/L was taken? All methods should be improved and present clearly the experimental details.
Thank you for checking. The error has been corrected.
- Why Swiss albino mice were used during in vivo experiments? If n=6 (number of animals in each group) is statistically enough to give the proper answer about the biological impact of MPP?
Swiss albino mice were more readily available for the development of the tests. Following the protocol approved by the animal ethics committee (CEUA), which considers 6 animals per group statistically acceptable for this type of assay.
The number of animals was chosen based on a variety of factors, including statistical, ethical, and practical considerations. This was done to ensure meaningful results while minimizing unnecessary animal use. The use of six animals per group was determined based on statistical considerations to ensure that the study has statistical power.
Below are some articles that we based our choice of the number of animals on:
- PMID: 24894401 PMCID: PMC4160094 DOI: 10.1167/iovs.14-14383 (Diabetic and non-diabetic mice of 5 to 7 animals)
- PMID:23255670, DOI: 4149/gpb_2012_039 (Diabetic rats n = 6)
- PMID: 23238787 PMCID: PMC3563947 DOI: 10.1007/s00125-012-2792-x (Diabetic rats n = 6)
- PMID: 33225469 DOI: 10.1096/fj.202001138RR (Mice n=6)
- Why during in vivo experiments doses equal to 100 and 200 mg/kg/bw were used? Also acarbose?
Firstly, a pilot test was conducted where doses of 100 and 200 mg/kg/body weight were established. Metformin at 100 mg/kg/body weight did not show in vivo activity.
- Why in some experiments quercetin was used as an additional control and what is the rationale for choosing presented concentration? The same doubt is for usage of doxorubicine (it is not related to the main topic of the study). Why metformin or acarbose were not chosen?
Metformin and acarbose are not standard compounds used to determine antiglycation activity in vitro, likely due to their low activity. Doxorubicin is a cytotoxic molecule that causes cell death, and this standard was used to demonstrate that MPP is not cytotoxic.
- Why the effect on glycation was not studied with the samples collected from in vivo experiments?
The in vitro antiglycation assay was performed, and the HbA1c of the mice was measured. AGEs were not measured due to a lack of access to kits at the time of collection because of the pandemic.
- Why the effect of MMP on differently originated a-glucosidase was not studied more deeply with molecular docking? The mechanism of enzyme inhibition has not been presented (despite the fact that the Authors have the tools).Why human originated a-glucosidase was not studied?
Performing docking studies would indeed greatly help in better understanding the difference between the two types of α-glucosidase enzymes described in the manuscript. Through docking, we could further analyze human α-glucosidase because our work only has authorization from the animal ethics committee and not yet from humans. We agree on the importance of unraveling the mechanism of action of MPP for future studies.
- Discussion of the results is VERY simple! It should present more detailed information!!! The bioavailability issue should be presented since it will have a great impact even on the binding with serum albumin.
The question of bioavailability was inserted as described below:
Table S2
Pharmacokinetic parameters of MPP
|
Physical Chemical Properties |
Result |
Parameters |
|
|
Molecular weight |
238,28g/mol |
< 500 |
|
|
Number of heavy atoms |
18 |
||
|
Aroma number. heavy atoms |
12 |
||
|
Number of rotating titles |
4 |
≤ 10. |
|
|
Number H-bond acceptors |
2 |
≤ 10; |
|
|
Number H-bond donors |
0 |
≤ 5; |
|
|
TPSA |
26,30Å |
≤ 140 Å |
|
|
Csp3 fraction |
0.06 |
||
|
Log P o/w |
3.27 |
≤ 5 |
|
|
Log S (SILICOS-IT) |
Moderately soluble |
||
|
Gastrointestinal Absorption |
High |
||
|
BBB permeant |
Yes |
||
|
P-gp substrate |
No |
||
|
Bioavailability score |
55% |
||
To predict the pharmacokinetics of MPP, the web tool SwissADME was used (http://www.swissadme.ch). MPP demonstrated compliance with Lipinski's Rule of Five and Veber's Rule, which enables its oral use [54]. Additionally, the compound showed a moderate to high tendency to dissolve in organic solvents, with moderate solubility in water (Table S2). These properties suggest the potential of MPP to cross the blood-brain barrier, but they may also influence its bioavailability, estimated at 55%, due to gastrointestinal absorption and protein binding (Figure 3). However, the compound's lipophilicity may reduce its oral absorption. To overcome this challenge, future studies will seek to develop a nanoparticle formulation aiming to enhance its effects without increasing the risk of toxicity.
12.The potential mechanism matching the effect of MPP on glucose uptake should be explained with more details. From my point of view, the residual attempts to discuss the obtained results with the literature data do not correspond to the level of the reviewed article.
Future research will be developed to explain the possible mechanism of action of MPP.
Reviewer 4 Report
Comments and Suggestions for Authors
The manuscript with the title "Anti-diabetic activities and GC-MS analysis of 4-Methoxychalcone" is an important part of a study that seeks to find a new anti-diabetic drug. The manuscript is well-written, very interesting scientifically, but also with a great practical probability. I would have some recommendations:
- In the Introduction: to be completed with recent data from the specialized literature about 4-Methoxychalcone: e.g. other synthesis methods, therapeutic actions, mechanisms of action, especially in diabetes, if the antidiabetic, antihyperglycemic action, toxicity, etc. have been studied.
- "Discussions" chapter - to compare your results with those of other authors.
- Bibliography: to be arranged and completed with the one related to 4-Methoxychalcone
Author Response
- The manuscript with the title "Anti-diabetic activities and GC-MS analysis of 4-Methoxychalcone" is an important part of a study that seeks to find a new anti-diabetic drug. The manuscript is well-written, very interesting scientifically, but also with a great practical probability. I would have some recommendations: to be completed with recent data from the specialized literature about 4-Methoxychalcone: e.g. other synthesis methods, therapeutic actions, mechanisms of action, especially in diabetes, if the antidiabetic, antihyperglycemic action, toxicity, etc. have been studied. - "Discussions" chapter - to compare your results with those of other authors. - Bibliography: to be arranged and completed with the one related to 4-Methoxychalcone
More references have been added to the text to improve the work:
Khan, A.; Khan, A.; Farooq, U.; Taha, M.; Shah, A. A. S.; Halim, A. S.; Akram, A.; Khan, Z. M.; Jan, K. A.; Al-Harrasi, A. Oxindole-based chalcones: synthesis and their activity against glycation of proteins. Med Chem Res, 2019, 28, 900–906.
D'Souza, A.J.; Topp, E.M. Release from polymeric prodrugs: linkages and their degradation. J Pharm Sci. 2004, 93,8,1962-79.
Karthikeyan, S.; Thirunarayanan, A.; Shano, L.B.; Hemamalini, A.; Sundaramoorthy, A.; Mangaiyarkarasi, R.; Abu, N.; Ganesan, S.; Chinnathambi, S.; Pandian, G.N. Chalcone derivatives' interaction with human serum albumin and cyclooxygenase-2. RSC Adv. 2024,17,14,2835-2849.
Rodrigues, G.S.; Avelino, J. A.; Siqueira, A. L. N.; Ramos, L. F. P.; Santos, G. B. O uso de softwares livres em aula prática sobre filtros moleculares de biodisponibilidade oral de fármacos. Química Nova, 2021, 44,8,1036–1044.
Zhang, J.; Cao, L.; Sun, Y.; Qing, D.-G.; Xu, X.-Q.; Wang, J.-C.; Si, J.-Y.; Li, N. The Regulatory Effects of Licochalcone A on the Intestinal Epithelium and Gut Microbiota in Murine Colitis. Molecules, 2021, 26, 4149.
Xiao, P.J.; Zeng, J.C.; Lin, P.; Tang, D.B.; Yuan, E.; Tu, Y.G.; Zhang, Q.F.; Chen, J.G.; Peng, D.Y.; Yin, Z.P. Chalcone-1-Deoxynojirimycin Heterozygote Reduced the Blood Glucose Concentration and Alleviated the Adverse Symptoms and Intestinal Flora Disorder of Diabetes Mellitus Rats. Molecules, 2022, 4, 27, 7583.
Reviewer 5 Report
Comments and Suggestions for Authors
Based on previous reports on the potential of chalcone and its derivatives for the treatment of diabetes, the authors also suggested to provide substantial evidence regarding the in vivo and in vitro antidiabetic activity of MPP. But in relation to the other derivatives' active effects, how does their effect stack up? More discussion needs to be added.
1. Please check sample abbreviations, especially in the formulas.
terephthalate (DT) and 4-Methoxychalcone (MPP).
2. The article has multiple occurrences of in vitro and in vivo; please italicize.
3. The results show that 4-Methoxychalcone has antidiabetic effects, so how does it compare to other chalcone derivatives? Please add relevant information.
4. In molecular docking, bovine serum proteins are used as receptor proteins. Why not directly use the crystal structure of human serum albumin?
5. The results of the non-oxidation pathway are inconsistent with the results shown; please check.
6. In the hemolysis experiment, the sample and blood cells were co-incubated for only 1h, which is too short of a time.
7. The results only show significant analysis with DNTG and do not infer that there is no significant difference in reducing blood glucose levels in the MPP200 group compared to the group treated with the same dose of metformin.
Comments on the Quality of English Language
Minor editing of English language required
Author Response
- Please check sample abbreviations, especially in the formulas. terephthalate (DT) and 4-Methoxychalcone (MPP).
Thank you for checking. The error has been corrected.
- The article has multiple occurrences of in vitro and in vivo; please italicize.
Thank you for checking. The error has been corrected.
- The results show that 4-Methoxychalcone has antidiabetic effects, so how does it compare to other chalcone derivatives? Please add relevant information.
Although other chalcone derivatives have shown antidiabetic activities below 200 mg/kg in preclinical studies, MPP is easily synthesized with a high degree of purity and has demonstrated safety in experiments. This work precedes another study where MPP is incorporated into a nanoformulation to increment its bioactivity.
- In molecular docking, bovine serum proteins are used as receptor proteins. Why not directly use the crystal structure of human serum albumin?
Why was the in vitro antiglycation test performed using bovine serum albumin.
- The results of the non-oxidation pathway are inconsistent with the results shown; please check.
Thank you for checking. The error has been corrected.
- In the hemolysis experiment, the sample and blood cells were co-incubated for only 1h, which is too short of a time.
Red blood cells are very sensitive to hemolysis, and in the protocol used, it is shown to incubate for a minimum of 30 minutes.
References:
Amin, K.; Dannenfelser, R.M. In vitro hemolysis: guidance for the pharmaceutical scientist. J Pharm Sci. 2006, 95,6,1173-6.
Jimenez P. C., Fortier S. C., Lotufo T. M. C., Pessoa C., Moraes M. E. A., De Moraes M. O., Costa-Lotufo L. V., Biological activity in extracts of ascidians (Tunicata, Ascidiacea) from the northeastern Brazilian coast. J. Exp. Mar. Bio. Ecol., 2003, 287, 93–101. https://doi.org/10.1016/S0022-0981(02)00499-9.
- The results only show significant analysis with DNTG and do not infer that there is no significant difference in reducing blood glucose levels in the MPP200 group compared to the group treated with the same dose of metformin.
The results show a similar activity at the same dosage used, as shown in the figure 7.
Round 2
Reviewer 1 Report
Comments and Suggestions for Authors
The authors have made a reasonable effort to address reviewer's comments.
Author Response
The entire text of the article was reviewed by a native English speaker. All possible grammar or typing errors have been corrected.

Reviewer 3 Report
Comments and Suggestions for Authors
I have read the Authors’ answer – they answered some of my doubts, but still many things are not clear. Speaking frankly, the answers are quite sloppy. I do not understand why they see the novelty in HS-GC-MS analysis, whereas the main hero of the manuscript - 4-methoxychalcone – is broadly available. Therefore more than half of the presented material does not contain any novelty. What is more, the Authors said that in the manuscript is presented fragment of data collected during realization of the project - I suggest to present here more data from this project instead of chemical synthesis, analysis and isolation of the 4-methoxychalcone – more information about the development of a nanoformulation aimed to increment bioactivity of 4-methoxychalcone.
In my opinion they did not explain why they used human lung fibroblast cells (MRC-5), not correlated with diabetes, as a cell line model used for searching compound with antidiabetic potential. Also there is no explanation why such long incubation was chosen for cytotoxic study.
Figure 4 -on X-axis is presented concentration within the range 0.6-20 microM, whereas for hemolysis 420 microM was used (it is not clear that as for the highest noncytotoxic?). Present the hemolytic activity for studied range of concentrations) and effect on metabolic activity at 420microM.
Explain with details presented in the answers what means “Metformin at 100 mg/kg/body weight did not show in vivo activity.”, since it is used as a therapeutic in humans treatment of diabetes.
There is no explanations “Why during in vivo experiments doses equal to 100 and 200 mg/kg/bw were used? Also acarbose?”. No data has been presented.
I suggest to get rid of data showing the doxorubicin or quercetin biological activity, since they are not related in any way to the 4-methoxychalcone.
I insist on addition of studies of the effect of MMP on differently originated a-glucosidases (including human originated a-glucosidase) with molecular docking (there are some programs available for free). The mechanism of enzyme inhibition also should be presented, since the manuscript presents rather collection of different data collected from different simple experiments, which are rather use for screening (not revealing of molecular mechanism).
In case of “bioavailability comment” I suspected to obtain a few sentences presented in the manuscript, not the raw data taken from specification.
I still insist on the research of the literature and explanation of the potential mechanism matching the effect of MPP on glucose uptake(comment about future study does not satisfy me). From my point of view, the residual attempts to discuss the obtained results with the literature data do not correspond to the level of the reviewed article.
In summary, I suggest the major revision.
Author Response
I would like to thank the reviewer for the comments that raise very pertinent questions with the aim of improving the quality of the material presented. The article does bring some new information about the 4-methoxychalcone molecule, even though it is a well-known molecule in the literature. As already reported, the article is part of a broader work that developed a nanoformulation with the molecule and will be published in the future, due to the amount of experimental data, it could not be included in a single article. Even with these considerations, we believe that the information presented in this article, such as the details of the methodology for quantifying the molecule by GC-MS and new infomations about hipoglycemic activities are relevants and has enough novelty for publication in a journal with the profile of AppliedChemm and which should be a widely cited article in the future.
Regarding the doubt regarding the use of the MRC-5 line, we emphasize that this fibroblast line is one of the most used lines when wanting to evaluate the cytotoxicity of new substances. The lineage is a fibroblast of human lung origin and is very sensitive to the toxicity of different substances. Generally, toxicities are assessed within 24, 48 and 72 hours. We carried out the evaluation at three times, which were presented in figure 4 and we consider these times adequate for the purpose of the experiment.
Regarding the hemolytic test, the maximum concentration tested was the same as that of the cytotoxicity test (20 microM). This has now been corrected in the figure caption and in the text.
Regarding the tests carried out in vivo on Swiss mice, different doses of treatments were previously tested in pilot tests with reduced numbers of animals before carrying out the definitive experiment in order to establish the dose for the final experimente. In relation to metformin, doses lower than 200 mg/kg did not show results in reducing blood glucose levels in both tests carried out. For this reason, the dose of 200mg/kg was chosen, carried out and presented in the article.
Regarding comparative data in assays with standards such as doxorubicin and quercetin, we emphasize that these are well-known standards and recognized as cytotoxicity and antioxidant standards, respectively. We believe it is worth remaining with the data presented so that readers can make comparisons and have an idea of ​​the potency of the activity of the substance tested in this experiments.
Regarding the addition of molecular doking data with glycosidases of different origins, it is indeed a possibility and a good tip, however it would generate new figures and increase the size of the article and the number of figures. We assess that at the moment the in vitro tests carried out already evaluate the specificity of the molecule in at least two types of glucosidases as evaluated in the article and presented in figure 5.
Regarding data on pharmacokinetics, this will be better explored in the second article on the molecule that will be published together with the nanoformulation produced.
Regarding the mechanisms of chalcones involved in the hypoglycemic effect, the last paragraph of the material and methods section discusses the possible mechanisms involved. In the text a suggestion of the 4-MMP mechanism was suggested and a short text was now added with the probable hypoglycemic mechanisms of chalcones according with previus literature.
The entire text of the article was reviewed by a native English speaker. All possible grammar or typing errors have been corrected.

Reviewer 5 Report
Comments and Suggestions for Authors
-
Author Response

(The authors gave the same response as above.)

Round 3
Reviewer 3 Report
Comments and Suggestions for Authors
I have read the Authors' answers - they answered some comments. I stand by my comments. In order not to prolong the editing process, I choose a minor revision dropping the decision-making to the Editor.